# TOX Expression in Mycosis Fungoides and Sezary Syndrome

**DOI:** 10.3390/diagnostics12071582

**Published:** 2022-06-29

**Authors:** Alessandro Pileri, Martina Cavicchi, Clara Bertuzzi, Simona Righi, Corrado Zengarini, Elena Sabattini, Giovanna Roncador, Claudio Agostinelli

**Affiliations:** 1Dermatology Unit, IRCCS Policlinico Sant’Orsola, 40138 Bologna, Italy; corrado.zengarini@yahoo.it; 2Dermatology Unit, Department of Experimental Diagnostic and Specialty Medicine, University of Bologna, 40138 Bologna, Italy; 3Department of Dermatology, University of Modena and Reggio Emilia, 41121 Modena, Italy; martycavicchi@gmail.com; 4Hemathopathology Unit, IRCCS Policlinico Sant’Orsola, 40138 Bologna, Italy; clara.bertuzzi2@unibo.it (C.B.); simona.righi5@unibo.it (S.R.); elena.sabattini@aosp.bo.it (E.S.); claudio.agostinelli@unibo.it (C.A.); 5Haematopathology Unit, Department of Experimental Diagnostic and Specialty Medicine, S. Orsola-Malpighi Hospital, University of Bologna, 40138 Bologna, Italy; 6Monoclonal Antibodies Unit, Spanish National Cancer Research Institute (CNIO), 28029 Madrid, Spain; groncador@cnio.es

**Keywords:** TOX, mycosis fungoides, Sezary syndrome

## Abstract

Mycosis fungoides (MF) and Sezary syndrome (SS) are the two most common type of cutaneous T-cell lymphoma (CTCL). Currently, no markers can be clearly related to prognosis or to differential diagnosis between early stages and inflammatory benign diseases (IBD). The thymocyte selection-associated high mobility group box factor (TOX), has been proposed as a possible marker in differential diagnosis between early CTCL stages and IBD. Recently TOX has been related to prognosis. We aimed to investigate whether TOX may be a diagnostic or prognostic marker. MF and SS biopsies between 2010 and 2020 were retrieved. New tissues slides were stained with an anti-TOX antibody, (Clone NAN448B). On each slide, 5 fields were examined at high magnification (400×), to evaluate the percentage of marker-positivity in a quantitative way. Thirty-six patients (12 females and 24 males) and 48 biopsies were collected. Nine patients had multiple biopsies. TOX expression in MF/SS cases showed an increase from early to advanced phases. TOX was not regarded as a prognostic marker due to the absence of significant changes by comparing early MF cases with reactive conditions. TOX statistical significance increased in patients alive with disease and in those dead of disease (*p* = 0.013 and = 0.0005, respectively) as compared with patients in complete remission. Our results show that TOX should be regarded more as a prognostic than a diagnostic marker.

## 1. Introduction

Mycosis fungoides (MF) and Sezary syndrome (SS) are the two most common cutaneous T-cell lymphoma (CTCL). MF and SS commonly affect the male gender in the fifth decade of life. Different studies have estimated that CTCL incidence has been tripled over the last three decades, with a number of new cases raised from 0.19 case/100,000 people/year to 0.46 among Dutch population [1,2,3,4,5]. 

In a recent publication Dobos et al observed a change in incidence of different CTCL subtypes [6], while other studies have detected areas featuring high CTCL incidence [7,8]; Identification of geographic clustering and regions spared by cutaneous T-cell lymphoma in Texas using 2 distinct cancer registries [4,5,9]. MF/SS pathogenesis and progression are still a matter of debate. Airborne toxins and potential carcinogenetic substances have been investigated as possible trigger of the disease with no clear correlation [10]. Infective agent susceptibility such as Staphylococcus aureus is thought to be involved in MF/SS pathogenesis and progression [11]. Genetic aberration along with cytokinc and inflammatory cell changes have been proposed as a putative candidate [12,13,14,15,16,17,18,19,20,21,22,23,24,25,26,27] sometimes providing contradictory results due to the small sample size or the use of different technique of investigation (immunohistochemistry and/or molecular biology analysis). Genetic aberration involving cell cycle and proliferation, chromosomal instability, and DNA repair have been observed in MF/SS [28]. Those genes are involved in different pathways such as epigenetic and/or chromatin regulation, TCR and T-cell/ cytokine signalling, Jak/signal transducer and activator of transcription (STAT), and phosphoinositide 3-kinases (PI3K)/protein kinase B (Akt) and NF-kB pathway, configuring that a complex mutational landascape may be related to MF/SS pathogenesis [27,29,30]. Also microenvironment changes from early to advanced CTCL phases have been proposed in the pathogenesis of MF and SS.

Currently, none of the available immunohistochemical markers are capable of help in the differential diagnosis between MF/SS and inflammatory diseases nor are there genetic aberrations or immunophenotypical profiles that can be definitely related to MF/SS pathogenesis or progression [31,32]. The thymocyte selection-associated high mobility group box factor (TOX), is part of the superfamily of high mobility group box proteins that act as modulators of gene expression by modifying the chromatin structure [33,34]. TOX is involved in the maturation of T lymphocytes in the thymus, guiding their transition from double positive (CD4+ CD8+) to CD4+ [34]. TOX expression is absent after lymphocyte migration from the thymus to the peripheral blood. Over the last few years several studies have evaluated the TOX expression in various neoplasms demonstrating that TOX is involved in the differentiation of “tumour-specific” T cells and in the “exhaustion” (anergy) of CD8+ T lymphocytes [35,36,37]. It has been hypothesised that TOX may promote the formation of exhausted T-cells akin to what happens after the stimulation of immune checkpoints such as PD-1 or CTLA-4 [38,39]; and it may play a role in cancer pathogenesis and progression. Some studies have evaluated TOX role in MF/SS providing contrasting results on its potential role as a diagnostic and prognostic marker [31,32,40,41,42,43,44,45,46]. The present study aims to evaluate TOX role of in MF/SS as a possible diagnostic marker by comparing its expression in MF/SS and in inflammatory skin diseases, and to assess whether TOX may also have a prognostic role.

## 2. Materials and Methods

### 2.1. Biopsies Evaluation and Immunohistochemical Analysis

The study was approved by the local IRB (MF.Tox.Isto20). MF/SS biopsies examined between 01/01/2010 and 12/12/2020 at the Dermatology and Haematopathology Units of Bologna University were collected. For each patient, clinical data concerning age, sex, clinical stage [47], treatment before and after study biopsy and follow-up were obtained (Table 1). For mycosis fungoides and Sezary syndrome diagnosis, EORTC-CLTF criteria were followed [47] and all Sezary syndrome patients were regarded as stage IVA1. Sections were deparaffinized and subjected to rehydration through Histoclear and alcohol at increasing gradations and stained with Hematoxylin-Eosin and May-Grunwald Giemsa. Immunohistochemistry was performed on 2 µm thick formalin-fixed paraffin-embedded (FFPE) whole tumour sections using a Dako Autostainer Link48 (Dako Agilent, Glostrup, Denmark), after antigen retrieval. For antigen retrieval, slides were incubated in a PT-Link for 5 min at 92 °C in the EnVision Flex Target retrieval solution High pH K 8004, Dako, Glostrup, Denmark) [48]. For diagnostic purposes all the examined cases had been stained with the following antibodies: CD1a, CD2, CD3, CD4, CD5, CD7, CD8, CD30, CD20, TIA, Granzyme B, Perforin, Ki-67/MIB-1. Three independent pathologists (CB, ES, CA) reviewed all the cases (reactive and MF/SS diagnosis). The latter were confirmed by basing the diagnosis on the criteria of the Fourth Edition of the WHO Classification of Haematopoietic and Lymphoid Tumours. New sections were incubated with an anti-TOX antibody, (Clone NAN448B) at a concentration of 10 μg/mL, at room temperature for 30 min, washed with phosphate buffer containing 2% BSA and incubated with Avidin Biotin Colorimetric Kit (Vector Lab, Burlingame, CA, USA) in the presence of Fast red chromogen, and then counterstained with haematoxylin. The slides were reanalysed as above described for reactive and MS/SS. The skin biopsies were analysed with the objective lens of 40 × /0.75 Olympus UPlanFL (Olympus, Italy), corresponding to an area of 0.237 mm^2^. On each slide, 5 fields were examined at high magnification (400×), to evaluate the percentage of marker-positive cells as follows: score 1 (Figure 1): <10% positive cells, score 2 (Figure 2): 10–50% positive cells, score 3 (Figure 3): >50% positive cells.

### 2.2. Statistical Analysis

Statistical analysis was performed using non-parametric tests because the available data had neither a normal distribution evaluated with Kolmogorov-Smirnov test nor homogeneity of variances verified with Levene test. TOX expression was evaluated in four groups to establish a possible diagnostic marker: reactive cases, eMF (stages IA, IB with disease in patch-plaque), late MF (stages IIB, IIIA, IIIB, IVA) and SS. For the purpose of evaluating TOX changes among the groups, the Kruskal-Wallis test was used for unpaired data followed by Dunn’s post hoc test for multiple comparisons. Regarding the potential prognostic role of TOX, the marker expression was evaluated in the group of patients alive disease-free (A−), versus alive with disease (A+), and deceased due to the disease (DOD). For such an analysis the Kruskal-Wallis test was used for unpaired data followed by the Dunn’s Post hoc Test for multiple comparisons.

## 3. Results

### 3.1. Patient Characteristics

A total of 48 biopsies corresponding to 36 patients (12 females and 24 males) were selected. Nine patients (four females and five males) had multiple biopsies, six showed a progression to an eMF, three developed tumours. Twenty-seven had a diagnosis of MF and nine of SS. Fourteen MF patients were characterized by an early stage with patchy-plaque disease at diagnosis (eMF: stages IA, IB), while 13 were in advanced stage with tumor-stage lesions or erythroderma (late MF: stages IIB, IIIA). The mean age of patients with eMF was 55, 4 ± 11.8 (range 36–70 years), those with late-stage MF had a mean age of 69.3 ± 15.1 (range 41–94 years) while in patients with SS the mean age was 57.2 ± 9.6 years (range 44–77). Of the 36 patients reviewed, seven were alive disease-free (A−), 16 were alive with disease (A+), and 13 had died (DOD). All the patients in early stage underwent skin-directed therapies such as phototherapy (PUVA or narrow band UVB) and application of high-potency topical steroid (i.e., dipropionate clobetasol). Eight out of 13 advanced-stage patients were treated with mono-chemotherapy (gemcitabine or pegylated liposomal doxorubicin) and six of them underwent to additional therapies, such as Brentuximab vedotin in four cases, or mogamulizumab in the remaining ones. In five instances radiotherapy was the preferred choice: in three instances limited-field radiotherapy was selected, while in two instances (#18 and #34) TSEBI and Bexarotene was scheduled. After the treatment, 10/14 stage IA/IB patients showed a disease complete remission (CR) and four experienced a partial remission (PR). In the advanced patients, six showed a stable disease after the first course of chemotherapy, while a PR was observed after brentuximab/mogamulizumab administration. In the patients treated with radiotherapy, a CR was observed after limited-field radiotherapy protocol, while two patients (No. 18 and 34) showed a downstage of the disease featuring patch and plaque lesion without the signs of tumour or blood involvement. The clinical characteristics of the patients are summarized in Table 1.

In the control group, ten patients (four males and six females) with inflammatory skin diseases were selected, four chronic eczema, two psoriasis, two lichen planus, and two granuloma annulare. The control group had a mean age of 64.2 ± 5.3 (range 55–73) and all patients were alive at the time.

The features of the control group are summarized in Table 2.

### 3.2. Tox Expression

In brief, TOX expression in MF/SS cases showed an increase from early to advanced phases. Indeed, the mean TOX value was 2.05 in early phases and 2.44 in advanced cases. The mean TOX value in reactive cases was 1.36. Statistical analysis showed no significant changes in TOX expression by comparing early MF cases with reactive conditions. However, the comparison between controls and late MF phases and SS showed a significant increase (mean 1.36) and late MF phases and SS (mean 2.44) in TOX expression in both diseases (*p* = 0.04 and *p* = 0.006, respectively, Figure 4).

The second part of the analysis (focused on the evaluation of the prognostic role of TOX) showed a statistically significant increase in marker expression in A+ patients (*p* = 0.013) and in DOD ones (*p* = 0.0005) as compared with A− patients (Figure 5).

Evaluating the expression of Tox in biopsies of A−, A+ and DOD patients considering separately the three groups of eMF, late stage MF and SS showed a maximum value of the marker in all cases with worse prognosis, regardless of the stage considered (Figure 6).

Descriptive analysis of the individual biopsies and the ten cases of patients of whom multiple biopsies were available showed that an increase in TOX expression related to a worse clinical outcome. A paradigmatic example are cases #21 and #25 that developed a transformed MF (one patient DOD and the other A+) featuring an increase in TOX expression which reached the maximum score. Interestingly, in two patients (#18 and #34) a decrease in TOX levels was observed after treatment with TSEBI and Bexarotene. The patients had a clinical downstage associated with a reduction in Tox value. In contrast, patients #14 and #36, who both died of disease, had high levels of Tox (Figure 7).

## 4. Discussion

The role of TOX as a possible MF-specific diagnostic marker has been evaluated in different studies [31,32,42,49]. Zhang et al. [49] first proposed TOX as a possible marker of differential diagnosis between CTCL and benign inflammatory conditions owing to a significant different expression. In their analysis Zhang et al. [49] observed a TOX higher expression in 21 MF cases by comparing with 21 reactive dermatitis. The authors speculated whether TOX expression in neoplastic T-cells may depend on a re-acquired ability to express TOX by MF cells after migration from the thymus rather than a loss of in the marker expression. Further studies showed contradictory results: Huang et al. [42] corroborated the findings of Zhang et al [49], while other groups observed opposite results ruling out the hypothesis of considering TOX as a MF-related marker [31,32,42,49]. Yu et al [40] observed an increased expression in advance-stage MF when compared with early lesions. In their analysis Tox expression was evaluated in 35 MF patients and compared with 10 controls featuring inflammatory skin diseases. Yu et al. [40] observed a significant marker expression comparing inflammatory skin cases, to MF/SS ones. Moreover, a stage-related increase in TOX levels was found analysing tumour (high expression) plaque (intermediate expression) and patch lesions (low expression). The authors supposed that TOX may play a crucial role in migration and proliferation of neoplastic T-cells. Higher TOX levels were observed in MF/SS if compared to reactive inflammatory diseases by Shrader et al. [32] corroborating the hypothesis that TOX may be regarded as a diagnostic marker. McGirt LY et al. [31] in their analysis described a similar expression in TOX in MF/SS and in parapsoriasis. The authors concluded that MF/SS and parapsoriasis may have similar pathogenetic mechanisms, while in our opinion TOX should not be regarded as a possible diagnostic marker. Recently, Dobos et al. [43] observed an increase in TOX levels in the peripheral blood, speculating as to whether TOX may be a helpful diagnostic marker in SS. Our analysis shows that TOX should not be regarded as a CTCL diagnostic marker, owing to the non-significant expression between early MF cases and different benign inflammatory diseases. Hence, we come to the same conclusion of McGirt LY et al. [31] concerning the reliability of TOX as a possible marker capable to discriminate inflammatory benign condition (featuring a low TOX expression) from MF/SS cases (high TOX expression). However, the increase in TOX expression comparing patch/plaque and tumour lesions along with changes in the marker’s expression in sequential biopsies of progressed cases warrants a comment. In accordance with Yu et al. [40] TOX expression was higher in advanced MF and in SS cases (*p*-value 0.04 and 0.006, respectively). Although TOX may not be helpful in the differential diagnosis between early MF and reactive conditions, its high expression in erythrodermic cases may be useful for differentiating erythrodermic MF- or SS from erythrodermic reactive cases such as atopic dermatitis or psoriasis. Such a differential diagnosis is important so as to avoid administering unsuitable treatments such as immunosuppressors to affected patients [50]. In the past two different groups have suggested that TOX may be considered a prognostic marker. Huang et al by analysing 113 MF/SS patients related an increased expression of TOX at gene level to an increased risk of disease-progression and disease-specific mortality [42]. A similar conclusion was reported later by Morimura et al [44]. The Japanese group described in their analysis an increase of Tox in TCD4+ lymphocytes localized at skin level in patients affected by MF/SS. Furthermore, Morimura et al. [44] described an increase of the marker levels as the disease progressed, a finding also observed in our cases. Indeed, in our analysis, the increase in TOX expression in advance-phase disease as well as the higher TOX expression in A+ and DOD patients when compared to A− ones corroborate the hypothesis that TOX may be regarded more as a prognostic than as a diagnostic marker [6,8,51,52]. Such evidence may be emphasized by the decrease in TOX levels in two patients (no 14 and 18) who had a relapse in early-stage patch after TSEBI+bexarotene treatment with a decreased expression in TOX levels. Hence, TOX may be helpful to monitor disease response after treatments and patients featuring high TOX levels in biopsy samples may be regarded as being prone to developing an aggressive disease, and consequently different therapeutic protocols such as new target molecules (i.e. brentuximab vedotin or mogamulizumab) may be considered. However, more cases are needed to corroborate our findings and to establish a possible cut-off between normal and high-risk patients. 

## Figures and Tables

**Figure 1 diagnostics-12-01582-f001:**
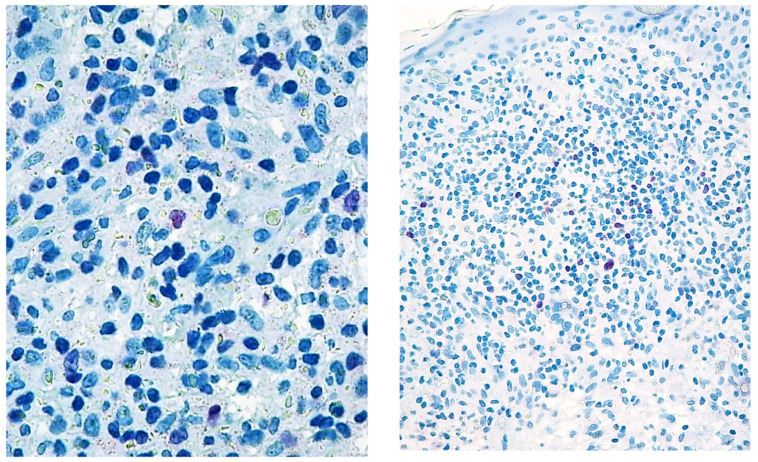
Paradigmatic example of score 1 of Tox expression in two cases.

**Figure 2 diagnostics-12-01582-f002:**
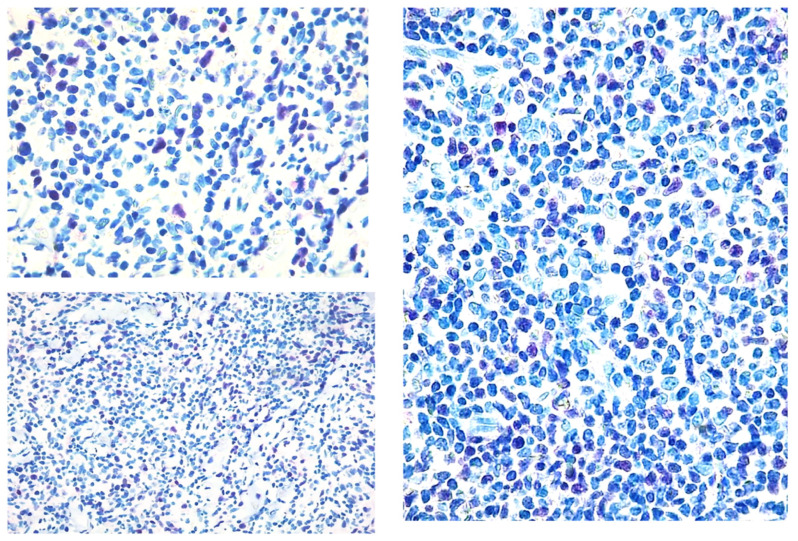
Paradigmatic example of score 2 of Tox expression in different cases.

**Figure 3 diagnostics-12-01582-f003:**
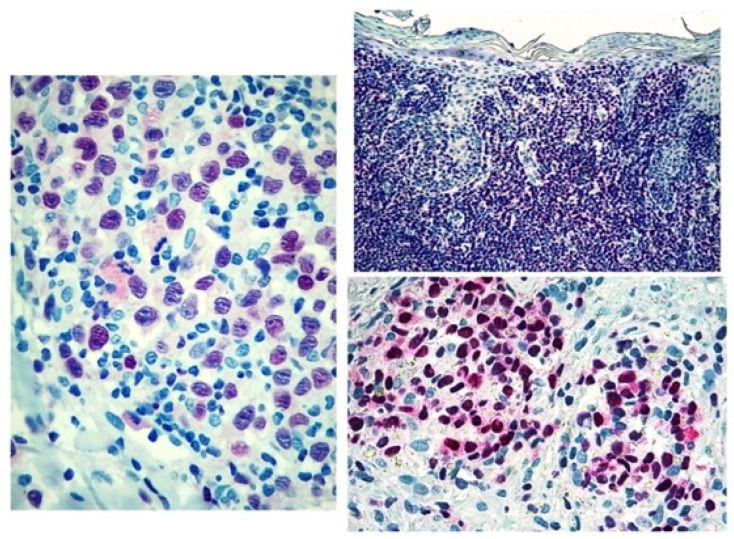
Score 3 Tox expression in early and advanced cases.

**Figure 4 diagnostics-12-01582-f004:**
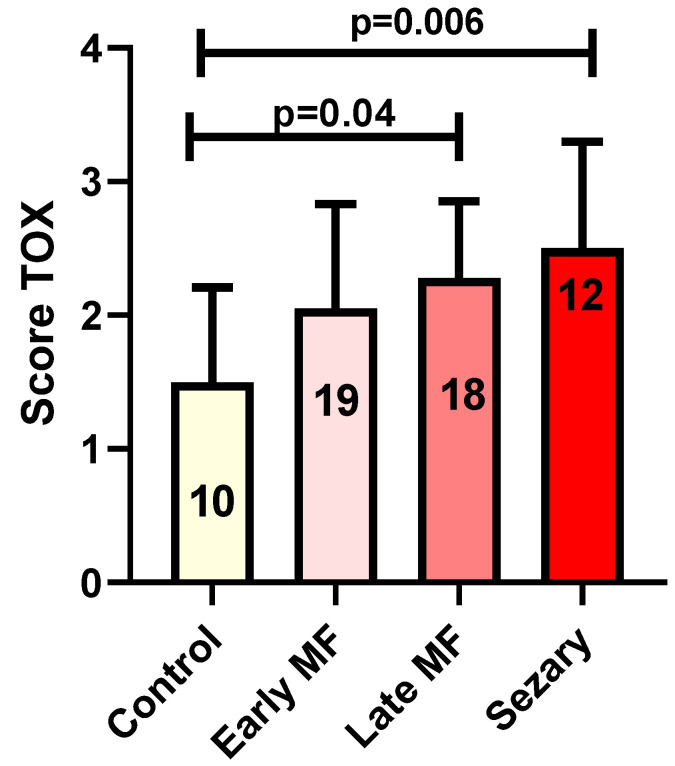
Tox expression in control group versus early MF, Late MF and Sézary Syndrome. In the columns: the number of biopsies evaluated in each group.

**Figure 5 diagnostics-12-01582-f005:**
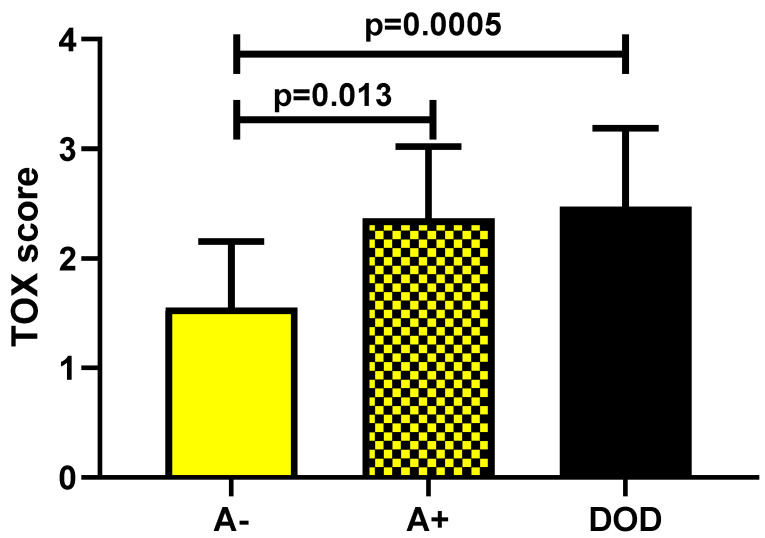
TOX expression in A− (alive without disease) versus A+ (alive with disease) and DOD (dead of disease) patients.

**Figure 6 diagnostics-12-01582-f006:**
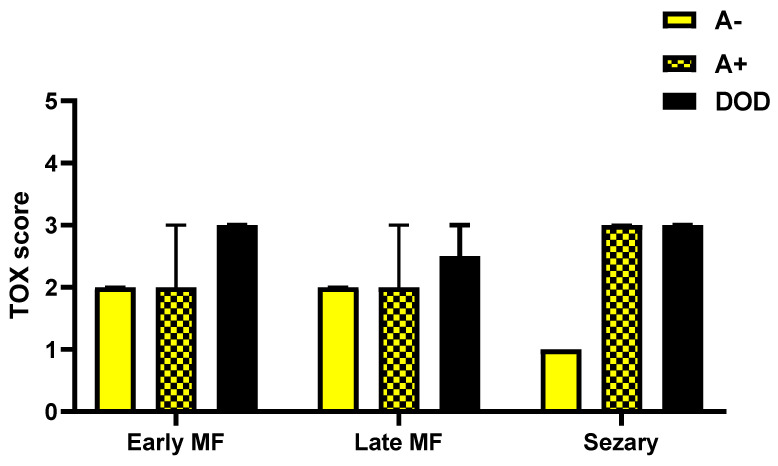
TOX expression in biopsies of A−, A+ and DOD patients considering separately the three groups of early MF, late stage MF and Sézary Syndrome.

**Figure 7 diagnostics-12-01582-f007:**
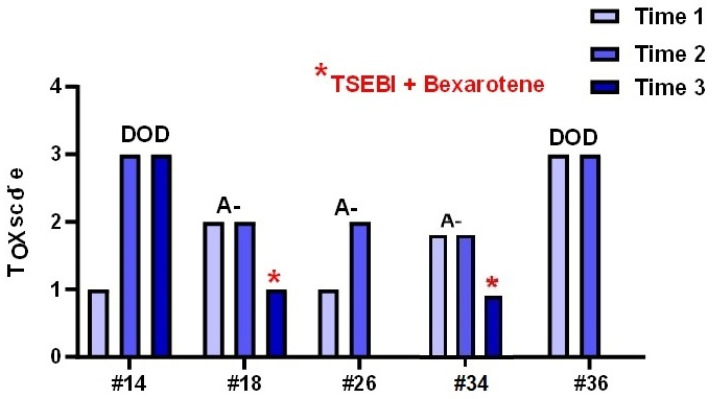
Tox expression in different biopsies of male patients performed during the clinical follow up of the disease.

**Table 1 diagnostics-12-01582-t001:** Clinical information of patients affected by MF/SS.

Patient ID	Age	Sex	Stage of Disease at Diagnosis	Maximum Stage	Alive with Disease (A+), Alive without Disease (A−), Dead of Disease (DOD)	Number of Biopsies Performed and Analyzed
#1	53	M	IA	IA	A−	1
#2	57	M	IA	IA	A−	1
#3	36	F	IA	IA	A+	1
#4	50	F	IA	IAb	A−	1
#5	64	M	IB	IBa	A+	1
#6	69	M	IB	IVA2	DOD	1
#7	69	F	IB	IIB	A+	2
#8	44	F	IB	IBb	A−	1
#9	62	M	IB	IBb	A+	1
#10	43	M	IB	IVA1	A+	1
#11	65	M	IB	IBb	DOD	1
#12	38	M	IB	IBb	A+	1
#13	70	M	IB	IBb	A+	1
#14	56	M	IB	IIB	DOD	3
#15	54	F	IIB	IIB	A+	2
#16	69	M	IIB	IIB	DOD	1
#17	79	F	IIB	IIB	DOD	1
#18	41	M	IIB	IIB	A−	3
#19	94	M	IIB	IIB	DOD	1
#20	87	M	IIB	IIB	DOD	1
#21	68	F	IIB	IIB	DOD	2
#22	60	M	IIB	IIB	A+	1
#23	54	F	IIB	IIB	DOD	1
#24	80	M	IIB	IIB	A+	1
#25	63	F	IIB	IIB	A+	2
#26	69	M	IIB	IIB	A−	2
#27	83	M	IIIA	IIIA	DOD	1
#28	68	M	IVA1	IVA2	DOD	1
#29	47	M	IVA1	IVA1	DOD	1
#30	60	F	IVA1	IVA1	A−	1
#31	44	M	IVA1	IVA2	A+	2
#32	70	M	IVA1	IVA1	A+	1
#33	62	F	IVA1	IVA1	A+	1
#34	62	F	IVA1	IVA1	A+	3
#35	56	M	IVA1	IVA1	A+	1
#36	46	M	IVA1	IVA2	DOD	2

1 A− (alive without disease) versus A+ (alive with disease) and DOD (dead of disease) patients.

**Table 2 diagnostics-12-01582-t002:** Clinical information of controls patients.

Pazient ID	Age	Sex	Diagnosis	Alive with Disease (A+), Alive without Disease (A−), Dead of Disease (DOD)	Number of Biopsies Performed and Analyzed
A	55	F	Chronic eczema	A−	1
B	60	M	Chronic eczema	A−	1
C	70	M	Granulomas annulare	A−	1
D	62	F	Psoriasis	A−	1
E	60	F	Lichen planus	A−	1
F	63	M	Granulomas annulare	A−	1
G	68	F	Chronic eczema	A−	1
H	73	F	Chronic eczema	A−	1
I	65	M	Psoriasis	A−	1
L	66	F	Lichen planus	A−	1

1 A− (alive without disease) versus A+ (alive with disease) and DOD (dead of disease) patients.

## Data Availability

Dataset has been internally generated.

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
