# Peer review of "TOX Expression in Mycosis Fungoides and Sezary Syndrome"

_diagnostics, 2022, doi:10.3390/diagnostics12071582_

Round 1

Reviewer 1 Report

I suggest to add a descriptive analysis of the clinical evolution ant treatment of all patieents with MF and SS included in the study

Author Response

First and foremost, we would thank the Reviewer for the constructive comments.

A descriptive analysis of the clinical evolution ant treatment of all patieents has been added in the results section and it is highlited in yellow

Reviewer 2 Report

In the current manuscript the authors evaluated the role of thymocyte selection-  associated high mobility group box factor (TOX) as a diagnostic and prognostic factor of in mycosis fungoides and Sezary syndrome. They noted that the TOX expression is related with disease progression rather than diagnosis.

Specific comments:

1. The graphical pictures are too many. The number of graphs should be less.

2. The microphotograph of TOX should be provided.

3. The statistical analysis part should be given

Author Response

First of all we would thank the Reviewer for the constructive comments.

The graphical pictures are too many. The number of graphs should be less.

R. We have reduced the number of graphical pictures

The microphotograph of TOX should be provided.

R. We have added TOX images

The statistical analysis part should be given.

R. We have changed statistial part accordingly.
